# Multi-human Interactive Talking Dataset

## Abstract

Existing studies on talking video generation have predominantly focused on single-person monologues or isolated facial animations, limiting their applicability to realistic multi-human interactions. To bridge this gap, we introduce MIT, a large-scale dataset specifically designed for multi-human talking video generation. To this end, we develop an automatic pipeline that collects and annotates multi-person conversational videos. The resulting dataset comprises 12 hours of high-resolution footage, each featuring two to four speakers, with fine-grained annotations of body poses and speech interactions. It captures natural conversational dynamics in multi-speaker scenario, offering a rich resource for studying interactive visual behaviors. To demonstrate the potential of MIT, we furthur propose CovOG, a baseline model for this novel task. It integrates a Multi-Human Pose Encoder (MPE) to handle varying numbers of speakers by aggregating individual pose embeddings, and an Interactive Audio Driver (IAD) to modulate head dynamics based on speaker-specific audio features. Together, these components showcase the feasibility and challenges of generating realistic multi-human talking videos, establishing MIT as a valuable benchmark for future research. *The code and data will be fully public available.*

## 1 Introduction

Recent advancements in human-centric video generation [25, 24] have markedly improved the synthesis of high-fidelity human videos. Among the most prominent research directions are pose-guided animation [5, 29, 16, 49], which enables fine-grained control over full-body movements, and audio-driven talking avatar generation [7, 10, 60], which focuses on producing accurate lip synchronization and expressive head motion conditioned on speech. Within the domain of audio-driven generation, substantial progress has been made in co-speech gesture synthesis [13] and talking head animation [38, 45]. The former seeks to align upper-body gestures with spoken content, while the latter aims to generate realistic facial expressions, head poses, and lip movements driven by audio input, thereby enhancing the expressiveness and naturalness of talking avatars. Despite these advances, existing methods predominantly focus on *single-person monologues* or *isolated facial regions*, lacking the capacity to model multi-speaker interactions. This limitation significantly constrains their applicability in realistic settings such as interviews, panel discussions, or films, where natural, multi-party conversations are essential.

In contrast to single-speaker scenarios, multi-speaker interactions involve complex dynamics, including turn-taking, fluid role transitions between speaking and listening, and non-verbal communicative behaviors such as eye contact and gesturing. Moreover, current datasets [9, 13] and generation frameworks [27, 26] are not designed to capture such multi-speaker conversational dynamics. Although recent work such as INFP [61] has taken initial steps toward interactive talking-head generation with multiple speakers, it remains restricted to facial animation alone. As a result, it fails to incorporate full-body behavioral cues critical for modeling realistic social interactions, thereby limiting both the quality and application of the generated content.

To advance beyond the limitations of single-speaker and facial-only generation, we define a new task, Multi-Human Talking Video Generation, which aims to synthesize realistic multi-person talking videos conditioned on reference images, body poses, and speech audio, as illustrated in Figure 1. Constructing a dataset suitable for this task is particularly challenging, as it requires the accurate extraction of multi-person conversational scenes, stabilization of camera motion, and

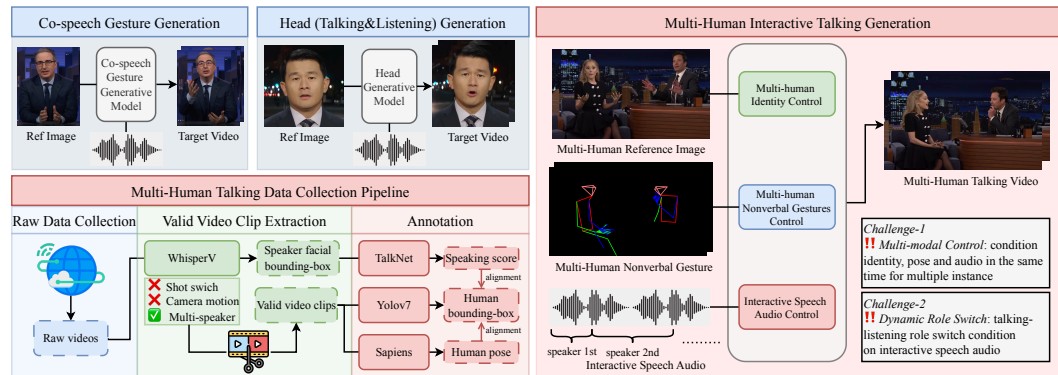

Figure 1: **Single Speaker Generation** *v.s.* **Mulit-human Interactive Talking Generation** and **Automatic Data Collection Pipeline**. The pipeline of existing tasks are shown in blue, Co-speech Gesture Generation [13, 26], and Talking or Listening Head Generation [10, 45]. In contrast, Multi-person Interactive Talking Generation enables dynamic speaker interactions by incorporating identity, interactive pose and audio control, as shown in red. And the automatic data collection is shown consisting of raw data collection, valid video clip extraction and annotation.

the removal of occlusions and post-production artifacts. In this paper, we propose an automatic data collection pipeline and use it to build a benchmark for this task. Specifically, we introduce the Multi-human Interactive Talking dataset(MIT), a fine-grained collection of 12 hours of multi-human videos featuring 2–4 speakers with diverse identities. This dataset includes multi-human pose annotations aligned with each speaker's speaking score label that indicates whether the human is speaking. Furthermore, we propose a baseline model designed for this task, namely CovOG: ConversationOriginal. Built on AnimateAnyone [1], CovOG integrates two key components: the Multi-Human Pose Encoder (MPE) and the Interactive Audio Driver (IAD). The MPE aggregates individual pose embeddings, allowing the model to accommodate a flexible number of human speakers. Meanwhile, the IAD dynamically refines speaker-specific head and pose features using an audio-driven speaking score, ensuring smooth and natural transitions between speaking and listening. Our work aims to lift audio-driven human-centric video generation to a more realistic setting, offering a significant contribution to the field.

To summarize, the contributions of this paper are:

- To the best of our knowledge, we first explore multi-human talking generation which lift exiting audio-driven video generation to a more realistic, universal setting.

- We develop an automatic data collection pipeline and construct the first dataset for multi-human talking video generation, featuring annotations of pose and speech interaction.

- We present a baseline model for this novel task, which supports a flexible number of human speakers and captures the dynamics of speech interactions. We further conduct extensive studies to benchmark our baseline against existing methods and analyze its performance.

## 2 RELATED WORK

### 2.1 HUMAN-CENTRIC VIDEO GENERATION MODEL

Recent advancements in diffusion models [35, 37, 14, 6, 52] have significantly enhanced video generation in terms of length, quality, and controllability. Stable Video Diffusion [4] employs latent diffusion to model video distributions within a latent space, enabling efficient and high-quality video synthesis. Furthermore, DiT-based models [31], such as CogVideoX [51] and MovieGen [36], improve video length and fidelity by diffusion transformers. Building on the advancements of these base models, human-centric video generation [25, 24] has garnered increasing attention due to its significant application potential. Text-driven models, such as Performer [21] and DirectorLLM [39], synthesize diverse human motions based on text prompts. Meanwhile, pose-based methods [11, 5, 29]

generate fine-grained controllable motions by leveraging pose sequences and reference images. Notably, AnimateAnyone [16] employs ControlNet [53] to maintain identity consistency throughout motion synthesis, while MagicAnimate [49] integrates an additional control branch to achieve better pose alignment.

## 2.2 Audio-Driven Character Animation

**Single Portrait Image Animation.** Single portrait image animation, which generates a talking or listening head from a given audio and portrait image, has recently gained significant attention. In talking head generation, various datasets [38, 9, 41] have been proposed. Notably, MEAD [47] focuses on emotion control, offering data across eight emotions with three intensity levels, while CelebV-HQ [59] provides diverse identities in realistic settings. Early approaches [33, 44, 54] relied on GAN-based models to improve lip synchronization. Recently, diffusion-based models [40, 20, 7, 10, 46] have significantly enhanced realism, consistency, and control ability. In listening head modeling, RLHG [56] first proposed ViCo dataset and built a sequential auto-encoder to generate non-verbal facial feedbacks given the speech audio and portrait image. Recent approaches [18, 30, 12, 27] have advanced reaction quality and controllability(*e.g.*, pose and text), by leveraging superior generative models(*e.g.*, VQ-VAE) and LLMs.

**Single-human Co-speech Generation.** Co-speech generation enhances single-head generation by incorporating nonverbal gestures, making the content more expressive. To facilitate research in this area, a high-quality dataset, SSGD [13], has been developed, providing co-speech video clips of 10 speakers along with pose annotations. Early approaches [34, 28, 60, 15] typically follow a two-stage pipeline: first, human poses are generated based on speech audio, and subsequently, pose-to-video methods (*e.g.*, AnimateAnyone [16]) are employed to synthesize co-speech gesture videos using a reference image. More recently, some studies have explored retrieval-based solutions for this task. Gesture video reenactment [58, 26] utilizes a short reference video clip (*e.g.*,, two minutes) to generate stylized gesture videos that align with novel speech inputs, resulting in more faithful and visually coherent outputs.

**Multi-human Conversation Generation.** Despite notable advancements in audio-driven single-human animation, it remains limited in capturing the richness of multi-human interactive conversations, which are more common and expressive in real-world applications (*e.g.*, movie dialogues, talk show interviews, and live streams). Recently, several studies [45, 57, 43] have explored interactive head generation, producing two talking-listening heads in a dyadic manner forming a conversation. Notably, INFP [61] introduced a large-scale dataset comprising extensive head-only conversational videos between two individuals and proposed an interactive motion guide to facilitate seamless talking-listening transitions. These approaches are constrained to generate only two individuals' head areas, as they fail to incorporate non-verbal contents such as eye contact, physical interaction, thereby restricting their applicability in more dynamic and natural conversational full-body interaction settings. Moreover, existing studies primarily focus on ideal turn-taking scenarios, where speakers alternate systematically, while challenges such as rapid role-switching and overlapping speech remain inadequately addressed. Existing methods fail to address multi-human talking generation in terms of full-body interactions and dynamic talking patterns, which requires specific models and datasets to capture multi-human interactive talking videos.

## 3 Multi-Human Interactive Talking Datset

We present a high-quality dataset for multi-human interactive talking video generation, comprising over 12 hours of high-resolution conversational clips with diverse interaction patterns and approximately 200 distinct identities. The dataset was constructed through a fully automated pipeline, facilitating future scale-up with minimal manual intervention. We provide a detailed description of this process in the following subsections, covering the data collection methodology (Section 3.1) and a analysis of interaction types and annotation statistics (Section 3.2).

### 3.1 Automatic Data Collection Pipeline

As illustrated in Figure 1, the data collection pipeline comprises three main stages: raw video collection, valid clip extraction, and multi-modal annotation. First, conversational videos are collected

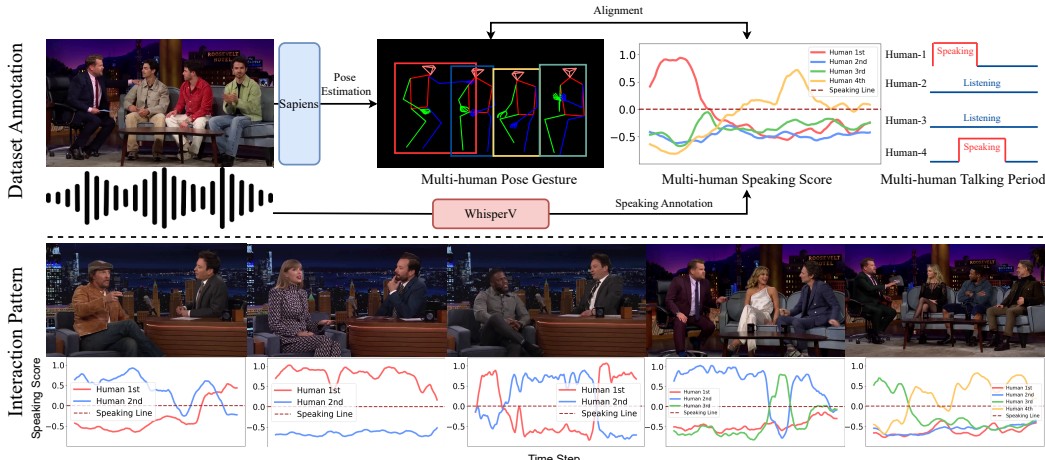

Figure 2: **Multi-human Interactive Talking Dataset.** Sapiens [23] and WhisperV [17] are used to annotate multi-human gesture and interactive speech respectively. MIT dataset captures rich conversation interaction pattens of multi-human, such as talking-listening, tune-talking, over-talking and other complex pattens.

from online platforms. However, most real-world videos undergo post-editing and include multiple shots from different perspectives (*e.g.*, close-up shots of faces and wide shots of the entire scene), which are unsuitable for current video generation models that require temporally consistent visual content. To address this, WhisperV [17] is adopted to segment videos into individual shots and to track facial trajectories of speakers within each shot. Clips featuring multiple active speakers within a single continuous shot are then extracted to preserve interactive dynamics. Finally, foundational perception models are employed to extract speaking scores, human poses, and bounding boxes. The bounding boxes serve as spatial anchors to align multi-modal signals, enabling consistent annotation for each individual speaker.

**Pose Annotation.** As part of the annotation process, 2D skeletal keypoints are extracted using Sapiens-2B [23] in the COCO133 [22] format. A subset of 59 keypoints is selected to represent the head, body, arms, legs, and hands, as illustrated in Figure 2. Specifically, only three keypoints are retained for the head to define its orientation, as finer facial expressions (e.g., lip movement, emotions) are primarily driven by audio. Notably, although the detected pose keypoints are pseudo-labels rather than manually annotated ground truth, they are obtained using a state-of-the-art pose estimation model, similar to SSGD [13]. This provides sufficient accuracy for generation tasks despite the absence of human supervision.

**Speaking Score.** In parallel, speaking scores are extracted using TalkNet [3], a model that performs speech activity detection. As illustrated in Figure 2, each individual is associated with a speaking score curve indicating periods of speech and silence. A score approaching 1 indicates active speaking, while a score nearing -1 corresponds to non-speaking states. The figure further illustrates how speaking scores reflect various interaction patterns: clear alternation between high and low scores indicates speaker turns; overlapping high scores across speakers correspond to simultaneous speech; and smooth transitions between high and low values capture speaking–listening dynamics.

**Pose–Speech Alignment.** After obtaining pose annotations and speaking scores—which are independently extracted and thus not inherently aligned—alignment is performed for each individual using human bounding boxes detected by YOLOv7. For each frame, pose annotations are assigned to the individual whose bounding box contains the highest number of keypoints. Similarly, each face track is matched to the individual whose bounding box most frequently overlaps with the facial bounding boxes across frames, leveraging the fact that face tracks are already aligned with speaking scores. By using the human bounding box as a shared spatial reference, both pose and speech annotations are consistently associated with the correct individual.

### 3.2 DATASET ANALYSIS

**Data Source.** Real-world videos often contain camera motion, occlusions, and post-editing artifacts, which are challenging to remove and typically require extensive manual intervention, such as region-

Table 1: **Existing Datasets** *v.s.* **MIT**. Compared to previous datasets that focus on single-person speech and isolated facial animation, our MIT dataset uniquely features multi-person talking videos with full-body interactions.

| Dataset | Num. | Area | Character | Pose | Speak | Res. | Total Len.(h) |
|---|---|---|---|---|---|---|---|
| SSGD [13] | One | Body | Speaking | ✓ | ✗ | 1920×1080 | 144 |
| HDFTD [55] | One | Head | Speaking | ✗ | ✗ | 512×512 | 16 |
| ViCo [56] | One | Head | Listening | ✗ | ✗ | 384×384 | 2 |
| RealTalk[12] | Two | Head | Interactive | ✗ | ✓ | 1280x720 | 115 |
| DyConv [61] | Two | Head | Interactive | ✗ | ✓ | 400×400 | 200 |
| MIT | Multi | Body | Interactive | ✓ | ✓ | 1920×1080 | 12 |

specific inpainting. To mitigate these issues while ensuring diverse and interactive multi-speaker scenarios, we curate classic and representative interview videos from two channels—*The Tonight Show*[1] and *The Late Late Show*[2]—as our data sources. These videos feature interactive multi-speaker scenarios that reflect real-world social behaviors, captured with static camera setups and minimal occlusions, making them well-suited for training models on interactive talking video generation. Despite the limited scene variety, the dataset features complex interactions and diverse identities, demonstrating its potential applicability to news, live broadcasting, and cinematic content.

**Interaction Pattern.** Multi-human interaction patterns constitute a critical yet challenging aspect of generating talking videos with multiple speakers, due to their inherent diversity and complexity. The most common pattern is turn-taking, where speakers alternate their roles, as explored in prior works [61] for interactive talking head. However, real-world conversations often exhibit more intricate dynamics, such as interruptions (over-talking), pauses, and rapid shifts between speaking and listening roles. Figure 2 illustrates the diverse interaction patterns captured in the MIT dataset, highlighting its suitability for advancing research in multi-human talking video generation.

**Dataset Statistics.** A comparison between MIT and existing datasets is presented in Table 1. MIT is the only dataset that features multi-human full-body interactions within conversational contexts. Although the total duration is limited to 12 hours, the automated data collection pipeline enables future scalability, compensating for this limitation.

**Quality of Data Annotations.** On a subset of 20 testing videos, we evaluate the automatic pose detections against human annotations and find that the pseudo ground truth is sufficiently accurate for our task. We also manually annotate the speaking–listening transition points (*i.e.*, the zero point of the speaking score) for each speaker, achieving an average temporal error below 0.1 second. Furthermore, we verify that pose–speaking alignments of all samples are correct.

## 4 BASELINE: COVOG

To tackle this task, we introduce CovOG, a tailored model built upon the single-person animation framework AnimateAnyone [16] which leverages Stable Diffusion [4] as base model and ensures identity consistency through ReferenceNet while incorporating conditional poses by embedding their features into the latent space via Pose Guider. Expanding on this foundation, CovOG integrates two key modules: the Multi-Human Pose Encoder (*i.e.*, Pose Guider/Adaptor) and the Interactive Audio Driver (IAD) as shown in Figure 3. The detail of each module is provided below.

### 4.1 NETWORK ARCHITECTURE

**Overview.** The overview of CovOG is shown in Figure 3 (a). Specifically, the multi-human pose embedding is incorporated into the multi-frame latent noise as pose control before being fed into DenoisingNet. Additionally, ReferenceNet is introduced for identity control using reference images, while IAD modules are incorporated to control the facial area based on speech audio.

---

[1]https://www.youtube.com/@fallontonight
[2]https://www.youtube.com/@TheLateLateShow

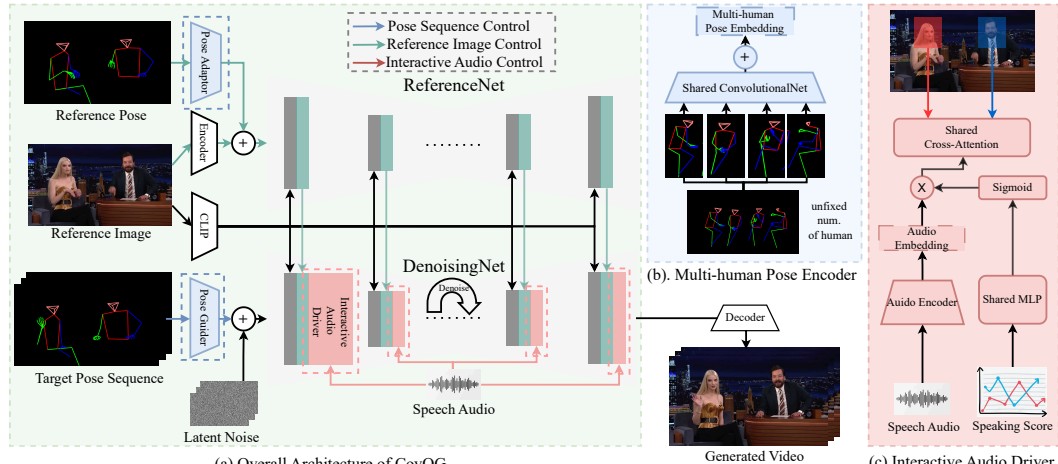

Figure 3: **Overview of proposed method CovOG.** (a) The overall architecure of CovOG. (b) Implement of Multi-human Pose Encoder used in Pose Adaptor and Pose Guider. (c) Implement of Interactive Audio Driver to capture the dynamic facial interaction between multiple speakers.

**Multi-human Pose Control.** To address multi-human pose control, we propose the Multi-human Pose Encoder (MPE) as Pose Guider as shown in Figure 3 (b). This module begins by utilizing instance masks to isolate individual human poses. Next, a shared convolutional network $\mathcal{F}_{\text{pose}}$ extracts features from each human pose $p_i$ separately. Finally, these features are aggregated to generate a unified embedding that comprehensively represents the poses of all individuals:

$$e_{\text{pose}} = \sum_{i=1}^{n} e_{\text{pose}}^i; \ e_{\text{pose}}^i = \mathcal{F}_{\text{pose}}(p_i), i = 1, 2, ..., n,  \tag{1}$$

where $e_{\text{pose}}^i \in \mathbb{R}^{f,c,w,h}$ stands for pose embedding of each human. This design is motivated by two key considerations. First, since pose control is independent for each individual, the model extracts and processes poses separately using a shared convolutional network, which promotes identity-invariant representations. Second, given that the number of individuals is variable, our design enhances robustness by allowing the network to manage pose of each human independently rather than being confined to a fixed number of individual.

**Multi-human Identity Control.** As the reference image contains multiple identities needed to be controlled, we propose a Pose Adaptor with the MPE architecture that extracts multi-human spatial cues. First, the reference pose is input into the Pose Adaptor to obtain a pose embedding. This embedding is then fused with the latent representation of the corresponding reference image and fed into ReferenceNet to provide spatial cues for each individual. This approach effectively accommodates variations in both the positions and the number of individuals across cases.

**Multi-human Interactive Audio Control.** In split of the complex patterns of interaction mentioned in Section 3, the speaking scores of individual speakers serve as a good indicator of the underlying interaction patterns. As shown in Figure 3 (c), we proposed the Interactive Audio Driver(IAD) to model the alignment between audio features and the corresponding lip movements and facial expressions. For $i$-th speaker, we use his speaking score $a^i \in \mathbb{R}^f$ to adjust the audio embedding $e_{\text{audio}} \in \mathbb{R}^{f,m,d}$. Subsequently, we employ the adjusted audio embedding $e_{\text{audio}}^i$ and hidden features $h_k$ from the DenoisingNet to perform a cross-attention $\mathcal{F}_{\text{audio}}$ using a facial mask:

$$h_{k+1} = h_k + \sum_{i=1}^{n} \mathcal{F}_{\text{audio}}(h_k, e_{\text{audio}}^i, \text{mask}_i); \ e_{\text{audio}}^i = e_{\text{audio}} \cdot \sigma(\text{MLP}(a^i)),  \tag{2}$$

where the parameters of this module are also shared across all speakers and $\text{mask}_i$ is obtained by the bounding box computed using three key head landmarks of human $i$. This design not only ensures that the model learns an identity-invariant alignment between audio and facial features, but also models the entire interactive process, thereby achieving a natural transition between listening and speaking. As shown in Figure 3 (a), the IAD module is inserted after each DenoisingNet block.

Table 2: **Quantitative Comparison and Ablation Study.** Experiments are conducted on the TonightShow for two-human scenarios and the LateLateShow for multi-human scenarios, under both easy and challenging test. The data from TonightShow consists of conversations with 2 speakers, while data from LateLateShow includes dialogues involving 2 to 4 speakers. Bold text indicates the best, while underlined text represents the second best.

| Method | Two Human | | | Multiple Human | | | All Test | | |
|---|---|---|---|---|---|---|---|---|---|
| | SSIM↑ | PSNR↑ | FVD↓ | SSIM↑ | PSNR↑ | FVD↓ | SSIM↑ | PSNR↑ | FVD↓ |
| *Comparison with Previous Methods* | | | | | | | | | |
| AnimateAnyone [16] | 0.60 | 18.98 | 322.08 | 0.64 | 19.96 | 353.11 | 0.62 | 19.47 | 337.60 |
| ControlSVD [48] | 0.31 | 13.46 | 1036.96 | - | - | - | - | - | - |
| CovOG | **0.62** | **19.16** | **306.01** | **0.66** | **20.21** | **308.68** | **0.64** | **19.69** | **307.35** |
| *Ablation Study* | | | | | | | | | |
| CovOG w/o MPE | 0.60 | 18.88 | 317.41 | 0.65 | 20.00 | 330.50 | 0.63 | 19.44 | 323.96 |
| CovOG w/o IAD | 0.61 | 19.06 | 313.69 | 0.65 | 19.86 | 347.92 | 0.63 | 19.46 | 330.80 |

## 5 EXPERIMENT

### 5.1 DATASETS AND EVALUATION METRICS

**Datasets.** In our experiment, we first split the test set from the MIT datasets, which consists of approximately 200 easy cases and 200 challenging cases sourced from both the TonightShow and the LateLateShow. The easy cases feature identities present in the training set but with novel pose and audio control parameters, whereas the challenging cases involve entirely unseen control signals to represent real application.

**Evaluation Metrics.** To qualitatively analyze model performance, we utilize Structured Similarity (SSIM), Peak Signal-to-Noise Ratio (PSNR) and Frechet Inception Distance (FVD) to evaluate the quality of generated samples. Unlike single-person talking head scenarios, lip alignment cannot be reliably assessed using LIPS [8] in our setting, as multi-person interactions involve both speaking and listening roles, often with side-facing views that LIPS is not designed to handle. How to effectively evaluate lip synchronization in such interactive contexts remains an open problem. To address this limitation, we complement our evaluation with user studies for visual-audio alignment.

### 5.2 IMPLEMENTATION

We pretrain our model following the two-stage paradigm proposed in AnimateAnyone [16], initializing it with weights from [1]. The model is trained on the entire training set, encompassing videos with varying numbers of speakers. The first stage and the second stage all comprised 30,000 steps with a resolution of 640×384, frame number of 15 and a batch size of 4 on 4 NVIDIA A6000 GPUs. The Pose Adaptor is integrated into the first stage and remains fixed in the second stage, while Interactive Audio Driver is incorporated into the second stage with the motion module. During inference, similar to Hallo2 [10], we utilize the final six frames from the previous inference as motion frames, incorporating them as the initial six frames of the subsequent inference while keeping them fixed to ensure the continuity and smoothness of generation. In addition, we obtained audio embedding using Wav2Vec [2].

### 5.3 COMPARISON

**Quantitative Evaluation.** We compare CovOG with two representative controllable video generation baselines: AnimateAnyone [16] and ControlSVD [48]. While more recent methods have emerged [32], we select these two due to their simplicity and broad representativeness, which allow for clearer comparisons. To ensure fairness, AnimateAnyone follows the same inference setup as CovOG. For ControlSVD, we use pose embeddings as input to ControlNet, initialize from the first frame, and generate videos autoregressively. As shown in Table 2, CovOG consistently outperforms both baselines across all metrics. AnimateAnyone struggles with multi-person scenarios, as its encoder jointly drives all subjects, while CovOG's MPE models each person independently and aggregates their effects. Moreover, lacking audio control, AnimateAnyone produces random facial

motions, whereas CovOG's IAD leverages personalized audio embeddings to enhance head dynamics and ensure audio-visual alignment. ControlSVD suffers from autoregressive error accumulation, leading to degraded quality over time, while CovOG maintains stability throughout generation.

**User Study.** We conduct a user study to evaluate character consistency, background consistency, audio-visual alignment, and overall visual quality. Seven participants rated 10 randomly selected samples per method on a 1–5 scale(higher is better), based on the reference image and speaking score. As shown in Table 3, CovOG outperforms other methods across all criteria, indicating superior control alignment and visual quality.

Table 3: **User Study**. 'CC', 'BC', and 'AV-Align' denote 'character', 'background consistency', and 'audio-visual alignment', respectively. 'Visual' indicates overall video quality.

| Method | CC↑ | BC↑ | AV-Align↑ | Visual↑ |
|---|---|---|---|---|
| *Comparison with Previous Methods* | | | | |
| AnimateAnyone [16] | 2.81 | 3.83 | 2.66 | 2.64 |
| ControlSVD [48] | 2.57 | 1.86 | 1.86 | 1.57 |
| **CovOG** | **2.93** | **4.11** | **3.22** | **3.34** |
| *Ablation Study* | | | | |
| CovOG w/o MPE | 2.64 | 3.55 | 2.79 | 2.5 |
| CovOG w/o IAD | 2.84 | 3.91 | 2.66 | 2.81 |

**Cross-modal Experiment.** To evaluate the generalization and practical applicability of our method, we conducted a cross-modal experiment. Specifically, we randomly selected 20 test cases by combining an identity image, a pose sequence, and corresponding speech audio from two different source videos, while ensuring that they involve the same number of speakers. Since ground-truth videos are unavailable for these cross-modal combinations, we employ VBench [19] to assess the generated results in terms of temporal consistency and visual quality, as shown in Table 4. The results demonstrate that CovOG achieves superior generalization both temporally and spatially.

## 5.4 Ablation Study

As shown in Table 2, removing either MPE or IAD leads to a clear drop in performance across all metrics. The absence of MPE results in the most significant decline, as torso control—essential for multi-person pose generation—heavily impacts visual quality. Without IAD, the model lacks sufficient control signals, causing unnatural head movements due to the absence of audio guidance. User study results in Table 3 further confirm these findings: character

Table 4: **Cross-modal Experiment**. 'SC', 'BC', 'AQ', and 'IQ' denote 'subject consistency', 'background consistency', 'aesthetic quality', and 'imaging quality', respectively.

| Method | SC↑ | BC↑ | AQ↑ | IQ↑ |
|---|---|---|---|---|
| AnimateAnyone [16] | 0.945 | 0.952 | 0.530 | 0.564 |
| CovOG | **0.952** | **0.959** | **0.542** | **0.603** |

and background consistency degrade without MPE, while audio-visual alignment suffers notably without IAD. These results validate the complementary roles of MPE for multi-person pose control and IAD for audio-driven facial synchronization.

## 5.5 Visualization Analysis

**Qualitative Evaluation.** We conduct qualitative evaluations on the MIT test set, as illustrated in Figure 4, where the first row presents relatively simple cases and the second row includes more challenging ones. The red and blue bounding boxes indicate the speaker and listener, respectively. Both methods produce plausible gestures. However, AnimateAnyone tends to generate an **averaged face** for both speakers and listeners. For instance, the listener's mouth remains static, and the speaker exhibits only limited lip movement. In comparison, CovOG shows a higher degree of interactivity and closer alignment with the ground truth. The speaker appears more engaged in speech, while the listener displays responsive expressions such as laughter. This may be attributed to CovOG's use of speaking scores to estimate speaking status, enabling adaptive facial expression generation. For example, when the input audio contains both speech and laughter, the model produces synchronized lip movements for the speaker and reactive expressions for the listener.

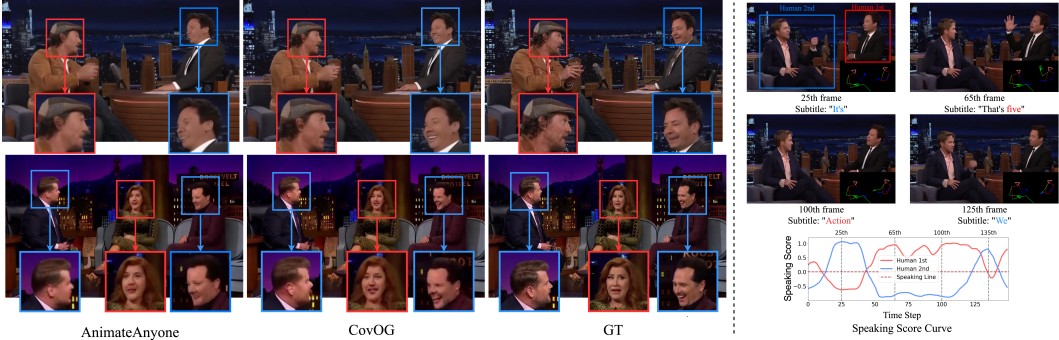

Figure 4: **Qualitative Comparison** and **Interaction Visualization**. Left: The red box indicates the speaker, and the blue box indicates the listener. Compared to AnimateAnyone, CovOG achieves superior lip synchronization for speakers and generates more natural, context-aware responses for listeners. Right: Visualization of the alignment with speaking scores, audio (*i.e.*, subtitles), and pose.

**Interaction Visualization.** We present the interaction visualization in the result generated by our CovOG, as shown in Figure 4. The speaking score curve indicates a turn-taking dialogue between two individuals. Key frames with their corresponding subtitles and the pose condition are displayed, with pronounced words highlighted in matching colors as in the speakings score curve. The results demonstrate that CovOG effectively aligns audio with lips and facial expressions for both speaker and listener, achieving natural interaction dynamics and strong audio-visual synchronization.

### 5.6 CHALLENGES IN MULTI-HUMAN TALKING SCENARIOS

Here, we outline the key challenges unique to multi-human talking scenarios in comparison to traditional talking-head and co-speech generation, and discuss the limitations of existing methods.

**Multi-huamn Interaction Modeling.** In a conversation, a person switches rapidly between speaking and listening, requiring the model to capture both the transitions and their dynamics. During speaking, accurate lip–audio synchronization is crucial, while during listening, the model only needs to produce natural, context-appropriate reactions. This difference in audio-visual patterns between speaking and listening poses a major challenge for generating realistic interactive speech.

**Side-Face Speech Alignment and Identity Consistency.** In multi-person conversational scenarios, speakers frequently turn their heads to engage with others, resulting in side-face appearances during speech. Accurately modeling lip movements in such cases remains challenging, as most talking head generation methods are primarily optimized for frontal views [42]. Furthermore, large rotational movements of the head and upper body pose challenges to maintaining visual consistency, particularly in facial features.

**Limitation of Existing Methods.** As discussed above, existing models face limitations in addressing these challenges. Moreover, talking-head methods are not designed to model full-body interactions, while co-speech models are often difficult to extend to multi-person scenarios. For instance, most recent work, TANGO [26] requires a two-minute reference video to construct an interactive audio–frame graph, which is impractical in multi-person conversations where audio–frame pairs are sparse. This sparsity hinders the feasibility to retrieve keyframes, leading to performance degradation.

### 6 CONCLUSION

In this paper, we introduce the Multi-human Interactive Talking (MIT) dataset, the first large-scale benchmark for multi-person talking video generation. To demonstrate its utility, we propose CovOG, a baseline model that integrates pose and audio cues to generate natural multi-human talking videos. We hope this dataset fosters further research in more challenging human-centric video generation.

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

## 7 CHECKLIST

### 7.1 THE USE OF LARGE LANGUAGE MODELS

In our work, LLMs are used for following aspects:

- Using an LLM to help with paper writing. We use GPT5 to help optimize language, correct grammar and write LaTeX table code.
- Using an LLM as a research assistant. We use GPT5 to help search related works.
- Using an LLM in our methods and experiment. This is described in the paper.

### 7.2 ETHICS STATEMENT

# 8 TECHNICAL APPENDICES AND SUPPLEMENTARY MATERIAL

Please refer to the supplementary webpage for video results.

## 8.1 DISCUSSION ABOUT BASELINE MODELS

Most existing studies primarily focus on talking-head generation or co-speech gesture synthesis. However, extending these methods to multi-human talking video generation presents significant challenges. In the following discussion, we elaborate on these limitations to clarify why only two representative baseline models are selected for comparison in the experimental section.

**Interactive Audio.** Unlike monologue scenarios with a single speaker, the audio in our setting involves multiple speakers, introducing a fundamental challenge: the model must accurately align each speaker's speech to the corresponding character in the video. Directly adapting existing methods proves difficult, as many are built upon assumptions of speaker continuity or global coherence. Consequently, key design components in prior models, such as Hallo2 [10] and TANGO [26], become ineffective in multi-speaker contexts. Specifically, TANGO constructs a graph for each speaker using approximately two minutes of reference video, where each node represents a video frame paired with a corresponding audio clip. This design enables the model to retrieve keyframes from the graph and generate transitions using an architecture similar to AnimateAnyone [16]. While effective in single-speaker scenarios, this approach faces critical limitations in multi-speaker contexts. The one-to-one correspondence between frames and audio segments becomes less reliable, and the graph becomes inherently sparse due to interactive audio patterns. As a result, it fails to support effective keyframe retrieval in multi-human settings.

**Mutli-human Pose and Identity Control.** This still remains a highly challenging task in controllable video generation. Although some recent works have explored this problem [50], they do not support audio-driven lip synchronization and still apply the ControlNet [53] architecture.

Overall, since most recent related works mainly apply ControlNet architecture we select AnimateAnyone (ControlNet for SD)[16] and ControlSVD[48] as baseline models, as they respectively represent the most relevant paradigms in single-human audio-driven generation and multi-human pose-conditioned synthesis, making them sufficient for evaluating performance in our multi-human interactive setting.

## 8.2 DISCUSSION ON EVALUATION METRICS

We evaluate model performance using both frame-level image quality and overall video quality metrics with respect to the ground-truth video. In addition, we conduct user studies and cross-modal experiments to assess lip synchronization and human–background consistency. However, unlike previous works on talking-head generation and co-speech gesture synthesis, we do not report quantitative lip alignment metrics [8]. This is because existing lip-sync metrics typically assume a single active speaker, which does not apply to our setting involving multiple speakers and interactive audio. The *interleaved* nature of speech in multi-human conversations makes such evaluations unreliable. Designing appropriate metrics for evaluating lip synchronization in multi-human scenarios remains an open research problem.

## 8.3 FUTURE WORK AND POTENTIAL IMPACT

**Multi-human Talking Pose Generation.** Our dataset also facilitates the study of multi-human pose generation in conversational contexts—an underexplored yet meaningful task. It offers an opportunity to investigate how generative models can capture human social dynamics. From a psychological perspective, this line of research may not only inform model design but also provide computational insights into nonverbal communication and social behavior.

**Dataset Scale-up.** With the proposed automatic annotation pipeline, we aim to scale up the dataset to cover more diverse scenarios, such as movies, live streams, and news broadcasts. This expansion will enable broader applications and support research under more varied and realistic settings.

**Multi-view Talking Video Generation.** We also plan to extend the dataset to include multi-view recordings, incorporating both wide-angle full-body interactions and close-up talking-head shots, as

commonly found in post-edited videos. This enhancement enables the exploration of multi-human generation in a multi-view setting, which better reflects real-world scenarios. In practical applications, human conversations are often captured from multiple viewpoints, making it essential for generative models to handle view-dependent rendering and ensure spatial and temporal coherence across views.

