# OpenReview forum: "Multi-Human Interactive Talking Dataset"
_ICLR.cc/2026/Conference — Submitted to ICLR 2026_

### Official Review · Reviewer_iKXf · 2025-10-29

**Soundness:** 3
**Presentation:** 3
**Contribution:** 3
**Rating:** 4
**Confidence:** 4

**Summary:**

This paper introduces the Multi-human Interactive Talking (MIT) dataset, designed for multi-person talking video generation, and presents CovOG, a baseline model that integrates both pose and audio cues to produce natural and realistic multi-human talking videos. The resulting dataset comprises 12 hours of high-resolution footage, each featuring two to four speakers, with fine-grained annotations of body poses and speech interactions.

**Strengths:**

1.This work presents the first dataset specifically designed for multi-human talking video generation, addressing the scarcity of multi-human interactive data.
2. It develops an automatic data collection pipeline and construct the first dataset for multi-human talking video generation, featuring annotations of pose and speech interaction.
3. A baseline model is proposed for this task, which supports a flexible number of human speakers and captures the dynamics of speech interactions. We further conduct extensive studies to benchmark our baseline against existing methods and analyze its performance.

**Weaknesses:**

1. Although this paper is the first to propose a dataset for multi-human talking video generation, several recent works have also addressed this task, including:
[1] HunyuanVideo-Avatar: High-Fidelity Audio-Driven Human Animation for Multiple Characters
[2] Let Them Talk: Audio-Driven Multi-Person Conversational Video Generation
[3] Bind-Your-Avatar: Multi-Talking-Character Video Generation with Dynamic 3D-mask-based Embedding Router

2. Current video generation models typically require a large amount of training data. However, the proposed dataset contains only about 2 hours of video, which is relatively small and may limit its applicability. Moreover, the videos are sourced from only two channels, resulting in limited diversity.
3. Most talking head generation methods employ evaluation metrics such as Sync-C and Sync-D (as introduced in[4]) to measure the synchronization between audio and lip movements. However, this paper does not include any evaluation of audio–lip synchronization, which makes it difficult to quantitatively assess the alignment quality of the generated videos.
[4]Out of time: automated lip sync in the wild.
4. It is unclear whether the proposed method supports single-person talking head generation. If it does, a comparison with other single-person talking head methods should be provided.
5. As far as I am aware, TalkNet’s scores can be highly unreliable when the speaker’s speech intervals are short. Additionally, the presence of background noise in the video may lead to evaluation errors. I would appreciate it if the authors could elaborate on how their data annotation pipeline addresses these issues.

**Questions:**

See weaknesses.

---

### Official Review · Reviewer_jgMV · 2025-10-31

**Soundness:** 3
**Presentation:** 2
**Contribution:** 2
**Rating:** 4
**Confidence:** 3

**Summary:**

The paper introduces a multi-human interactive talking video dataset and provides a first benchmark for this complex task. The paper provides a first baseline implementation, CovOG which utilizes pose + audio to generate multi-human videos.

**Strengths:**

The problem of multi-human interaction is highly relevant and challenging. Providing high-quality datasets for this task is crucial. To proposed baseline is simple yet effective.

**Weaknesses:**

The authors note that “How to effectively evaluate lip synchronization in such interactive contexts remains an open problem” (L353-354) and instead opt for a user study. While a user study is useful, I would argue for a dataset an accompanying is crucial. My concern here is that the dataset might be released prematurely without a good benchmark that measures how well the audio and video align, i.e. how well the speaker (lip movement, correct person)  and the audio are in agreement. I wonder if something like LIPINC [1] could be utilized for this.

I find utilizing 2D keypoints as the only conditioning (besides the reference image) limiting - For example, I wonder if the authors tried to extract SMPL poses from the videos?

Can the actors elaborate on why they not directly utilize the speaker scores but instead learn a projection in their interactive audio driver? It seems like the goal is to act as a form of Gating (Figure 3 - c) which can be achieved directly from the scores.

Can the authors elaborate why the generated human motion looks so smooth, both for animate anyone and their baseline? This is particularly noticeable in the face region. I wonder if this stems from the stable diffusion VAE? I believe their relatively low resolution (640x384) + the VAE play a role in this. Furthermore, there are visible background artifacts, which probably also stem from the image-based method. I wonder if the authors have tried video auto-encoders, i.e. the one from WAN to obtain more temporally stable results.

Relevant related works: [2] + [3]

[1] EXPOSING LIP-SYNCING DEEPFAKES FROM MOUTH INCONSISTENCIES, ICME 2024

[2] Towards social artificial intelligence: Nonverbal social signal prediction in a triadic interaction, CVPR 2019

[3] A Large-Scale Dataset for Multi-Shot Human Speech Video Generation; NeurIPS DBT 2025

**Questions:**

In Figure 1, according to the description, the  blue boxes represent existing pipelines and the red boxes represent additional pipeline extensions (paper contributions). What is the green box for?

**Details Of Ethics Concerns:**

The dataset is made up of Youtube videos to which the authors presumably do not have the rights to. I assume that videos of only public figures have been scraped for this. However, the authors do not provide an Ethics statement but I believe they should (briefly) discuss the ethics of the dataset collection and provide potential avenues for data deletion, if requested by the rights owner.

---

### Official Review · Reviewer_bcxr · 2025-11-01

**Soundness:** 2
**Presentation:** 2
**Contribution:** 2
**Rating:** 0
**Confidence:** 5

**Summary:**

In this paper the authors proposed a Multi-human Interactive Talking (MIT) dataset, a benchmark for multi-person talking video generation

**Strengths:**

The community really need a high-quality large-scale Multi-human Interactive Talking dataset. Where gestures, expressions and other dynamic behaviors of the human subject involve in the talking MUST BE TEMPORALLY AND PHYSICALLY CONSISTENT. However, the results shown in the supplementary video lacks all these attributes,

**Weaknesses:**

The video result shown in the supplementary results are not temporally and physically consistent.
For examples
- the number of fingers and their shapes changing across frames. Flat palm frequently morphing into fist suddenly
- Object in the hand are changing across frames,
- Facial expressions are not consistent across frames

**Questions:**

This dataset need a thorough user study to validate the curated/generated dataset is visually acceptable or not, physically accurate or not.

---

> ### Author Response · Authors · 2025-11-13
> **About dataset quality**
>
> Dear Reviewer,
>
> Thank you for your time!
>
> I think there may be a misunderstanding: the videos in the supplementary material are the generation result of CovOG instead of the videos from our dataset. The video frame samples are shown in Figure 2 and if you would like to see the videos in the dataset, I can share an anonymous GitHub link to showcase.
>
> I hope this clarification would be helpful for you to evaluate our work.
>
> Best regards!

---

> > ### Comment · Reviewer_bcxr · 2025-11-28
> > **Post Rebuttal**
> >
> > My concerns remains, hence I am keeping my original score.

---

### Meta-Review · Area_Chair_Zs6R · 2026-01-02

**Summary:**

While reviewers acknowledged the importance of the problem and the potential value of a multi-human interactive talking dataset, several serious concerns were consistently raised regarding dataset quality, evaluation rigor, and ethical compliance. Most critically, one reviewer reported that the video results lack basic temporal and physical consistency, with visible artifacts such as inconsistent hand shapes, changing finger counts, unstable facial expressions, and object discontinuities, raising doubts about whether the dataset and baseline meet the standard required for a benchmark dataset. Other reviewers questioned whether the dataset is sufficiently mature, due to the absence of reliable quantitative metrics for audio–lip synchronization and speaker–audio alignment, and expressing concern that the dataset may be released prematurely without a robust evaluation protocol. Additional weaknesses include limited dataset scale and diversity (relatively few hours of data from a small number of sources), methodological limitations of the baseline (e.g., reliance on 2D keypoints, overly smooth motion, background artifacts), and unclear design choices in the interactive audio driver. Finally, multiple reviewers raised ethics and legal concerns, particularly around the use of scraped YouTube videos, lack of a clear ethics statement, and absence of discussion on consent, copyright, and data removal mechanisms.

**Reviewer Concerns:**

The authors did not effectively address the reviewers' comments. Some of the reviewers' comments have not been addressed.

**Reviewer Scores:**

As one reviewer did not satisfy the authors' response, and the authors did not provide response to the other reviewers, it is unlikely for the reviewers to raise their scores.

---

### Decision · Program_Chairs · 2026-01-26

Reject